# Repair of Iron Center Proteins—A Different Class of Hemerythrin-like Proteins

**DOI:** 10.3390/molecules27134051

**Published:** 2022-06-23

**Authors:** Liliana S. O. Silva, Pedro M. Matias, Célia V. Romão, Lígia M. Saraiva

**Affiliations:** 1Instituto Tecnologia Química e Biológica António Xavier, Universidade Nova de Lisboa, 2780-157 Oeiras, Portugal; lsilva@itqb.unl.pt (L.S.O.S.); matias@itqb.unl.pt (P.M.M.); cmromao@itqb.unl.pt (C.V.R.); 2iBET, Instituto de Biologia Experimental e Tecnológica, 2780-157 Oeiras, Portugal

**Keywords:** Repair of Iron Center proteins, hemerythrin, di-iron protein, iron-sulfur biogenesis, nitrosative stress

## Abstract

Repair of Iron Center proteins (RIC) form a family of di-iron proteins that are widely spread in the microbial world. RICs contain a binuclear nonheme iron site in a four-helix bundle fold, two basic features of hemerythrin-like proteins. In this work, we review the data on microbial RICs including how their genes are regulated and contribute to the survival of pathogenic bacteria. We gathered the currently available biochemical, spectroscopic and structural data on RICs with a particular focus on *Escherichia coli* RIC (also known as YtfE), which remains the best-studied protein with extensive biochemical characterization. Additionally, we present novel structural data for *Escherichia coli* YtfE harboring a di-manganese site and the protein’s affinity for this metal. The networking of protein interactions involving YtfE is also described and integrated into the proposed physiological role as an iron donor for reassembling of stress-damaged iron-sulfur centers.

## 1. Introduction

During the infection process, IFNγ-activated phagocytes of the host innate immune system release superoxide and nitric oxide, whose derivatives are known as reactive oxygen species (ROS) and reactive nitrogen species (RNS). The nitrosative products that result in the combination of ROS with RNS are harmful to pathogens as they damage several bacterial components such as DNA, lipids, and proteins, which causes disruption of key cellular functions [1,2]. In particular, RNS bind to iron in heme cofactors and iron−sulfur (Fe-S) clusters, disabling, among others, respiratory heme-copper oxidases and enzymes of the tricarboxylic acid cycle (TCA), such as the [4Fe-4S]^2+/1+^ containing aconitase and fumarase. Bacteria have several protecting systems against nitrosative stress, including detoxifying enzymes like hemoglobins, NO reductases, and nitroreductases. Moreover, bacteria can restore the function of impaired cellular components through little-known mechanisms. Repair of Iron Center proteins (RICs) form a widespread family of proteins that are highly expressed under stress conditions. In this work, we summarize the current knowledge on the function of these proteins, namely their relationship with Fe-S cluster repairing. We also provide a new structure of the *Escherichia coli* YtfE in which the iron binuclear center is replaced by manganese ions and discuss its impact on the overall structure.

## 2. Microbial RICs

RIC stands for Repair of Iron Center proteins. The first RIC protein was discovered in *Escherichia* (*E.*) *coli* and is encoded by *ytfE*, a gene that is highly induced under nitrosative stress conditions [3]. The strain inactivated in *ytfE* is more sensitive to NO and H_2_O_2_. Moreover, YtfE was shown to protect and repair Fe-S cluster proteins, namely aconitase and fumarase [4,5].

RIC proteins are broadly spread in the bacterial phyla (Proteobacteria, Bacteroidetes, Firmicutes, Actinobacteria, Acidobacteria) and in eukaryotic pathogens [6]. Studies on several microorganisms reinforced the proposal that RICs are able to repair Fe-S clusters injured by ROS and RNS. For example, cells of *Staphylococcus aureus*, *Neisseria gonorrhoeae*, and *Trichomonas vaginalis* lacking RIC and treated with NO and H_2_O_2_ have lower survival rates and decreased activity of aconitase and fumarase, but these activities are restored upon addition of *ric* [5,6,7]. Therefore, this widespread family of di-iron proteins was named Repair of Iron Center protein [8]. Herein, YtfE will refer to the *E. coli* protein and RIC to the homologues present in the other microorganisms.

YtfE contains two iron atoms per monomer and is isolated as a combination of monomers and dimers [4], each monomer having a molecular mass of 25 kDa. The amino acid sequence of YtfE contains an HHE–hemerythrin cation binding motif analogous to that present in hemerythrin proteins [9]. The other studied RIC proteins have similar molecular masses, comprise also two iron atoms per monomer, and share a high amino acid sequence identity with *E. coli* YtfE (Table 1). The sequence alignment of more than 100 RICs revealed highly conserved histidine and carboxylate residues, namely H84, H105, H129, E133, H160, and H204 (*E. coli* YtfE numbering) [6], which are involved in the coordination of the binuclear site.

RICs are also present in other microbes such as the fungal pathogen *Cryptococcus neoformans* and the protozoan *Trichomonas vaginalis*. *C. neoformans* encodes a RIC-like protein (CNA2870) with two apparent hemerythrin cation binding motifs, which gene is induced by NO [10]. *T. vaginalis* is the only organism known so far to contain two RIC paralogs designated as RIC1 and RIC2. *T. vaginalis ric1* and *ric2 that* are both induced by nitrosative and oxidative stress. However, only *T. vaginalis* RIC1 shares similarities with bacterial RICs concerning the spectroscopic features and Fe-S repair ability. RIC2 is not isolated with a di-iron center and apparently lacks the capacity to assemble the binuclear center. Interestingly, RIC2 contains a signal peptide and a leucine zipper motif and is a DNA binding protein. Subcellular localization studies showed that RIC1 is present in the cytoplasm while RIC2 occurs also in the nucleus of *T. vaginalis* [7].

## 3. RICs Are Important for Pathogen Survival

RICs have been shown to contribute to survival in host cells of several human pathogens, such as *S. aureus, Haemophilus influenzae, Salmonella enterica Typhimurium*, and *Yersinia pseudotuberculosis* [11,12,13,14].

*S. aureus* Δ*ric* is more sensitive to oxidative stress and nitrosative stress and survives less than the wild type when infecting murine macrophages. Moreover, survival of *S. aureus* Δ*ric* is similar to the wild type when the mammalian cells are unable to produce ROS. Infection studies in the wax moth *Galleria mellonella* showed that *S. aureus* strains lacking *ric* are less virulent [11].

The *H. influenza* Δ*ric* mutant presents high sensitivity to NO and reduced ability to infect macrophages. This phenotype is canceled upon the addition of an inhibitor of the inducible NO synthase that impairs the NO production by macrophages [12].

Strains of *Salmonella enterica* and *Yersinia pseudotuberculosis* lacking *ric* present reduced survival during infection of the liver and spleen of mice and are less virulent to mice [13,14].

## 4. Regulation of RICs

The genomic organization of RIC encoding genes has a high degree of variation amongst microorganisms and these genes are not usually clustered with genes related to oxidative/nitrosative defenses. In the organisms so far studied, the *ric* genes are induced by nitrosative and/or oxidative stress [8]. In *E. coli*, YtfE is highly expressed in cells exposed to nitrosative stress and cells grown in nitrate and nitrite under anaerobic conditions [3,15]. In *S. enterica*, *S. aureus*, *T. vaginalis*, *N. gonorrhoeae*, *N. meningitidis*, *Ralstonia eutropha*, *Pseudomonas stutzeri* and *C. neoformans* the *ric* genes are upregulated by nitrosative stress. Additionally, *S. aureus* and *T. vaginalis ric*s are induced by oxidative stress [7,10,13,16,17,18,19,20,21,22]. *S. enterica*, *S. aureus*, and *Y. pestis ric*s are highly expressed during infection of macrophages and rats [11,23,24].

*E. coli*, *N. gonorrhoeae*, *N. meningitidis*, *S. enterica* and *Y. pseudotuberculosis ric*s are repressed by NsrR [6,14,16]. Additionally, *ytfE* is de-repressed in *E. coli* strains lacking the general regulators FNR and Fur. Since no recognized Fur-FNR binding sites are found in the promoter region of *ytfE*, the FNR/Fur regulation is proposed to occur indirectly through a not yet known mechanism [4,8]. *P. stutzeri ric* is positively regulated by DnrD regulator, and *R. eutropha ric* (also known as *norA*) is upregulated by NorR in response to NO [18,20]. *S. aureus ric* seems to be under a complex regulation that involves SrrAB, SarV, MgrA, and SarA. The two-component system SrrAB (which is activated by hypoxia and nitrosative stress) induces the *ric* expression when in the presence of NO [21]. SarV also controls positively the *ric* transcription [25]. MgrA, a member of the MarR family of regulators that is proposed to act as an oxidative stress sensor, negatively regulates *ric* [26]. Moreover, higher *ric* levels were measured in the Δ*sarA* strain indicating that *S. aureus* SarA represses the transcription of *ric*. In all cases, the binding of SarV, MrgA, and SarA to *ric* promoter remains to be shown. Additionally, it was proposed that regulation of *ric* by MrgA and SarA is indirect since the two transcription factors also repress the *sarV* transcription [25]. Figure 1 summarizes the current data on the regulation of RICs.

## 5. Biochemical and Spectroscopic Properties of RICs

UV-visible, EPR, resonance Raman, Mössbauer, and extended X-ray absorption fine structure (EXAFS) spectroscopic studies of *E. coli* YtfE were firstly used to analyze the binuclear iron center [4,28,29]. The UV-visible spectrum of YtfE displays a broad band at 360 nm characteristic of di-iron proteins, a feature that was also observed in RICs from *R. eutropha*, *S. aureus*, *N. gonorrhoeae* and *T. vaginalis* RIC1 [4,6,7,30].

The EPR spectrum of YtfE, acquired at ca. 15 K, exhibits *g*-values at ~2 (1.96, 1.92, and 1.88) typical of di-iron centers in the mixed-valence Fe^2+^-Fe^3+^ state, and similar spectra were also observed for *S. aureus* RIC and *T. vaginalis* RIC1 [4,6,7]. The Mössbauer spectrum of the *E. coli* ^57^Fe-YtfE proved that the di-iron center is in a mixed-valence state in the as-isolated protein. EPR-monitored redox titration allowed the determination of the two reduction potentials for the di-iron center: +260 and +110 mV, respectively for the Fe(III)Fe(III)-Fe(III)Fe(II) and Fe(III)Fe(II)-Fe(II)Fe(II) transitions [4]. Mössbauer data showed that the two iron atoms are antiferromagnetically coupled, with parameters consistent with the presence of a μ-oxo bridge in the iron center, and that one of the iron atoms is more labile [29]. Resonance Raman spectrum of as-isolated *E. coli* YtfE exhibits a band at 490 cm^−1^ which shifts to 478 cm^−1^ in ^18^O-labeled YtfE, indicating the presence of a μ-oxo group bridging the two iron atoms [28]. The EXAFS spectrum of *E. coli* YtfE is dominated by a single oscillation frequency due to ligands such as O and N. In the EXAFS structural model of *E. coli* YtfE, the two iron ions are coordinated by two histidine residues and two oxygen ligands at an average distance of ∼2 Å, and with a Fe-Fe distance of ∼3.6 Å that does not vary markedly between the ferric and fully reduced state. EXAFS of *E. coli* YtfE incubated with NO revealed the formation of a single {FeNO} species, a reaction that yields N_2_O. More recently, the access of NO to the di-iron center of YtfE was proposed to occur through a hydrophobic channel that connects the metal site to the N-terminal domain [31,32]. The ability of YtfE to bind NO is an expected observation as, in general, di-iron proteins bind small molecules like NO, O_2_, azide, thiocyanide, or chloride ions (e.g., [33]). As for bacterial methane oxygenases and ribonucleotide reductases [34,35,36], the *E. coli* YtfE NO reductase activity is also negligible [37]. More recently, *E. coli* YtfE was also described to reduce nitrite with the release of NO, with K_M_ for nitrite of ~250 μM and K_cat_ of ~35 min^−1^ [37]. This is a very slow reaction rate when compared with the values usually observed for heme and copper nitrite reductases (in the range of 200–800 s^−1^) [38].

## 6. Structural Properties of *E. coli* YtfE

### 6.1. RIC Cysteine Motif Does Not Modulate Its Oligomeric Form

*E. coli* YtfE and *R. eutropha* RIC were isolated as a mixture of monomers and dimers. In *R. eutropha*, reduction of RIC with DTT caused the loss of the dimeric form. Therefore, oligomerization was proposed to be related to the formation of disulfide bridges between the two cysteine residues of the highly conserved motif DfCCgG, which is present in the N-terminal domain in most bacterial RICs [30]. To address whether dimerization was linked to the oxidation of these residues, we determined the oligomerization state of a truncated version of *E. coli* YtfE that lacks the first 57 N-terminal residues and of site-directed mutated proteins where the cysteine residues C30 and C31 were replaced by alanine. Size exclusion chromatography analyses showed that the truncated YtfE is a monomer, as reported earlier [39]. The YtfE-C30A protein is eluted as a mixture of monomers and dimers, as is the case of the wild type protein. On the contrary, the C31A and C30AC31A double mutated proteins are monomers. Thus, we concluded that YtfE oligomerization depends solely on the C31 residue.

The possibility that the conserved motif DfCCgG was required for the formation of the di-iron center was also ruled out, as both the truncated version of YtfE and *N. gonorrhoeae* RIC, which lacks that N-terminal domain, still retain the di-iron center [6,39].

### 6.2. The YtfE Crystal Structure

The currently available X-ray structures of RICs are of that the wild type and mutated *E. coli* YtfE (PDB 7BHA, 5FNN, 5FNP, 5FNY) [31,40]. All structures could only be obtained when harboring the double mutation C30AC31 for reasons that remain unclear. YtfE has an L-shaped two-domain architecture: it exhibits an N-terminal ScdA_N domain connected through a highly flexible penta-peptide to the C-terminal four-helix bundle hemerythrin-like domain containing the di-iron site (Figure 2). The YtfE crystal structure contains two molecules in the asymmetric unit and all molecules show the two domains A and B, which correspond to the ScdA N-terminal domain and the hemerythrin-like domain, respectively. Different YtfE crystal structures were observed to fit in distinct space groups (*P*2_1_, *I*4_1_, and *P*2_1_ 2_1_ 2), indicating high variability in their inter-domain flexibility [31,40]. Domain A has a globular shape, is composed of 4 α-helices, and contains the conserved motif DfCCgG in the loop between helices 2 and 3. Domain B is composed of a 4-helix bundle fold and shares a similar structural topology with proteins such as hemerythrin, rubrerythrin, and (bacterio)ferritin, with a r.m.s.d. of ca. 3 Å between superimposed main-chain Cα atoms.

An extensive bioinformatics analysis comparing several hemerythrin-like protein structures showed that YtfE has a distinctive two-helix swap resulting in a left-handed four-helix bundle instead of the right-handed one typical of hemerythrins [41]. Thus, regarding the structural fold of the four-helix bundle, RIC proteins form a distinct cluster within the hemerythrin-like and ferritin-like protein families.

The di-iron site inserted in the four-helix bundle is coordinated by histidine residues through their N^ε2^ atoms: H84 and H204 bind Fe1, and H129 and H160 bind Fe2. The two Fe atoms are bridged by two bidentate glutamates, E133 and E208, and a μ-oxo ligand [31,40].

The YtfE fold forms two channels in the vicinity of the di-iron center. The longer channel (15–25 Å) is mainly formed by hydrophobic residues and connects the di-iron site to the ScdA N-terminal domain. The shorter hydrophilic channel (ca. 10 Å) includes solvent-exposed glutamate residues E125, E159, and E162. Residues E125 and E159 are highly conserved among RICs and are located at about 11.5 and 7 Å from the closest iron atom, respectively. E159 forms an H-bond with the H129 ligand and contributes to the stabilization of the di-iron site. This hydrophilic tunnel has a radius of ~2.3 Å and may accommodate small molecules or ions such as iron; thus, it is proposed to act either as a gate or cation trap [31,39,40].

### 6.3. YtfE-E159L Can Reassemble A Di-Manganese Site

Our previous studies indicated that mutation of E159 to leucine in YtfE affects the formation of the iron center [39]. The mutated protein no longer exhibits the EPR signal associated with the di-iron center, and the structure contains only one iron atom in the metal site (see below). The E159L mutation induces instability in the overall structure as judged by the higher atomic displacement parameter values when compared to the wild type YtfE. Moreover, the iron is weakly bound as deduced from iron release kinetics data [40].

During the optimization process of the YtfE-159L protein crystallization [40], we observed that when using an additive containing MnCl_2_ the two iron sites became occupied by manganese. The presence of a di-manganese center improved the overall quality of the crystals with the diffraction data measured at 1.9 Å resolution. Still, in this structure, designated as YtfE-159L^Mn^, coordination and ligand distances were maintained (Figure 3A, Table 2). Data collection and refinement statistics are summarized in Table 3.

These results led us to test the affinity of the wild type YtfE to manganese vs. iron. For this purpose, wild type YtfE was incubated with Mn^2+^ ions during the crystallization process and X-ray diffraction data were collected. Analysis of the anomalous difference maps obtained from data collected above the Mn and Fe K-absorption edges shows that the metal site contains in both positions a mixture of iron and manganese ions (Figure 3B,C): in position 2, which is closer to E159, manganese represented ca. 45% of the metal occupation, while position 1 retained almost all the iron, as only ca. 10% of manganese was present. No anomalous signal was observed before the Mn K-absorption edge (1.9016 Å) (data not shown). Therefore, iron in position 2 is more easily displaced by manganese than that in position 1 (Figure 3B,C). This observation is consistent with the Mössbauer data showing that in YtfE one of the iron atoms is more labile [29].

### 6.4. YtfE Binds Preferentially to Iron

Mn-catalases have their metals in the binuclear center bridged by carboxylates and inserted into a four-helix bundle fold, in a structure that shares similarities with that of di-iron sites in hemerythrin-like proteins [47]. Although Fe^2+^ and Mn^2+^ may compete for the same metal sites, their physiological functions are very different [48]. Hence it was of relevance to determine the affinity of the wild type YtfE to manganese and compare it with that of iron. This study was done by analyzing the quenching of the intrinsic fluorescence intensity [49] of YtfE upon the addition of Mn^2+^ or Fe^2+^, and determination of the binding constants for each metal. Both the apoproteins of the YtfE wild type and the YtfE-E159L variant exhibited intrinsic fluorescence emission spectra, with maxima intensities at ~326 nm. After the addition of Fe^2+^ or Mn^2+^ under reducing conditions in the absence of chelating agents, a decrease in the fluorescence intensity was observed, indicating the binding of the metal. Data showed that the wild type protein has a higher affinity to the two metals than the YtfE-E159L mutant. Moreover, the binding affinity constants of the wild type and mutated protein are significantly higher for iron than for manganese (*k_a_* = 6.5 ± 0.3 × 10^−6^ M (Fe^2+^) and 0.5 ± 0.2 × 10^−6^ M (Mn^2+^)).

These values clearly indicate that YtfE preferentially accommodates Fe, which is consistent with its physiological function.

## 7. The Physiological Function of RICs

### 7.1. Structural Features Related to the Iron Donation Properties of YtfE

*E. coli* YtfE can provide iron for the assembly of Fe-S centers in the apo-form of the spinach ferredoxin and *E. coli* IscU, when in reducing conditions and in the presence of IscS and L-cysteine. The *E. coli* YtfE dissociation constants for the ferric and ferrous iron are ~10^−27^ M and ~10^−13^ M, respectively [29]. Analysis of the *E. coli* YtfE structure shows that the negatively charged residues E125, E159, and E162 are near the di-iron center and the kinetics studies showed that both E125 and E159 control the iron donor properties [40]. We observed that the replacement of E125 and E159 by either hydrophobic or uncharged residues (leucine/asparagine) modifies the iron release properties. When compared to the wild type, the iron release rate of YtfE-E159L is approximately 7-fold higher and that of YtfE-E125L is ca. 8-fold lower [40]. Moreover, the two E125L/N mutants did not promote the assembly of Fe-S clusters in *E. coli* apo-IscU [29,40].

The effect of the replacement of E125 and E159 by leucine in the YtfE structure is depicted in (Figure 4). In YtfE-E125L, the hydrophilic channel is blocked by the side chain of leucine whereas the YtfE-E159L structure retains the iron channel present in the wild type [40].

Prior studies indicated that H129 is the only histidine ligand involved in the control of the iron donation capability of YtfE, as its replacement for a leucine residue produces a protein unable to restore the aconitase activity [39]. Interestingly, H129 forms an H-bond with E159 (H129*^Nδ1^*…E159*^Oε1^* 2.66 Å). Substitution of E159 by leucine caused the side chain of H129 to change its position towards the protein surface [40], thus impairing the coordination of the second iron atom. This might suggest that the movement of the H129 side chain is concomitant with the release of iron from the di-iron center through the hydrophilic channel. The structural modification of the histidine side chain would also allow the re-entry of an iron atom to rebuild the di-iron center. A similar mechanism has been proposed for the iron traffic in bacterioferritins, as these proteins also have a di-iron center (the ferroxidase site) inserted in a four-helix bundle that is linked to the protein surface through a hydrophilic channel of about 6 Å in length, which is known as the ferroxidase pore. This channel is suggested to be an alternate route for Fe^2+^ to access the ferroxidase site. Furthermore, the mechanism for iron inclusion contemplates conformational changes of histidine and glutamate residues in the ferroxidase site of bacterioferritin from *Azotobacter vinelandii*, *Pseudomonas aeruginosa*, and *Desulfovibrio desulfuricans* noticed in their X-ray structures [50,51,52,53].

In *E. coli* CyaY, IscA, SufA, and YtfE, have been considered as putative iron donors for the in vitro assembly of Fe-S clusters (Table 4).

*E. coli* CyaY is the homologue of human frataxin and a crucial protein for the assembly of Fe-S clusters of mitochondrial, cytoplasmic, and nuclear proteins, leading to its deficiency in the neurodegenerative disease Friedreich ataxia [54]. However, in *E. coli* and *S. enterica*, mutations in *cyaY* do not show a significant impact on Fe-S cluster metabolism [55,56]. More recently, *E. coli* CyaY was proposed to be a modulator protein of the IscS desulfurase activity, thus refuting its role as an iron donor [57,58,59].

Although both IscA and SufA bind iron, their role in the maturation of [4Fe-4S] clusters also remains elusive. In these proteins, the ferrous iron is weakly bound, as judged by the dissociation constants (Table 4) [60,61,62]. On the contrary, the ferrous iron dissociation constant of YtfE (∼10^−13^ M) is compatible with an iron donor role under physiological conditions.

**Table 4 molecules-27-04051-t004:** Dissociation constants of iron donors proposed for *E. coli* and YggX of *S. enterica*.

Protein	K_d_ Fe^3+^/K_d_ Fe^2+^ (M)	References
YtfE	10^−27^/10^−13^	[29]
CyaY	>10^−17^/10^−6^	[63,64]
IscA	10^−19^/nd *	[65]
SufA	nd/10^−5^	[66]
YggX	nd/10^−6^	[67]

* nd—not determined.

### 7.2. The Protein Interactions of E. coli YtfE

A search in an *E. coli* DNA library revealed that YtfE interacts with Dps (DNA-Binding Protein from Starved Cells) [68]. Dps is an iron-sequestering protein composed of 12 identical subunits forming a hollow shell surrounding a central cavity where up to ∼500 ferric iron atoms can be stored [69]. We observed that Dps inactivation increases the intracellular ROS levels in relation to cells lacking both *ytfE* and *dps* [68]. On the contrary, double and triple mutated strains lacking YtfE, catalases, and alkyl peroxidases showed no changes in ROS content. Moreover, other main iron storage proteins of *E. coli*, such as bacterioferritin and ferritin FtnA do not interact with YtfE.

Dps protects *E. coli* cells from oxidative stress generated by ROS formed through the Fenton reaction, which is triggered by free iron, and supports aerobic growth of cells lacking other ROS detoxification systems [70,71]. Studies with *Listeria innocua* Dps reported that iron is released from the Dps protein cage through 3-fold channels lined by three aspartate residues [72]. Furthermore, the iron release process is triggered by the reduction of ferric oxyhydroxide via reducing agents [72]. Therefore, the interaction of YtfE with Dps could serve to trap ROS released from YtfE di-iron core and/or provide iron needed for the reconstitution of the YtfE di-iron center, a hypothesis that requires further investigation.

*E. coli* YtfE also establishes interactions with proteins of the *isc* operon, which codes for proteins required for assembling Fe-S clusters, as shown by the bacterial two-hybrid system, molecular fluorescence complementation, and pull-down assays. In particular, it interacts with the scaffold protein IscU (that transiently assembles a Fe-S cluster) and with cysteine desulfurase IscS, which provides sulfur to the Isc system [40]. Our analysis of the genome of 20 high prevalent pathogens allowed concluding that RICs are found: (i) in bacteria that contain both Isc and Suf systems, as is the case of many bacteria of the Gammaproteobacteria and Clostridia classes; (ii) in microorganisms that contain only the Isc system, such as *Neisseria* spp. or the protozoan *T. vaginalis*; and (iii) in bacteria with the Suf system only, as is the case of bacteria of Bacilli (Table 5). Thus, the co-occurrence of RICs with proteins of the Isc and Suf systems may indicate a functional relationship.

Altogether the data so far gathered for YtfE such as the induction by NO stress, the interaction of YtfE with Dps and proteins of the iron-sulfur biogenesis system, the iron donor properties of YtfE, and the ability of YtfE to promote the formation of an iron-sulfur center in IscU and ferredoxin, all indicate that YtfE is a bacterial iron donor. On the contrary, the recently reported low value of nitrite reductase activity of *E. coli* YtfE was not yet shown to have physiological relevance [37].

## 8. Conclusions

RICs were first related to the protection of bacteria against NO due to their very high gene expression levels in cells exposed to NO and NO donors [3,8,13,15,17]. However, subsequent studies have shown that YtfE can donate iron for the assembly of Fe-S clusters with a dissociation constant for ferrous ions that is physiologically compatible with this function in vivo. Recent structural and biochemical data revealed that YtfE contains a hydrophilic channel that connects the di-iron site to the protein surface, with dimensions that allow iron traffic. Moreover, YtfE was found to interact with proteins that belong to Fe-S cluster assembly systems. Although more studies are still needed to fully understand the physiological conditions that trigger iron release from RIC and the systems that profit from this iron source, the data gathered is clearly consistent with a role of RICs as iron donor proteins for the assembly of iron-sulfur proteins, which is a novel function for hemerythrin-like proteins.

## 9. Materials and Methods

### 9.1. Crystallization and X-ray Diffraction Data Collection and Analysis

Proteins for crystallization were purified and concentrated to 20 mg/mL in buffer Tris-HCl 20 mM pH 7.5 with 150 mM NaCl. Crystallization experiments were done at room temperature using hanging drop 48-well VDX48 plates (Hampton Research) with 0.1 mL of reservoir in each well.

Crystals of *E. coli* YtfE^Mn^ and YtfE-E159L^Mn^ were prepared by the hanging drop vapor diffusion method using 1.0:0.8:0.2 µL mixtures of protein-reservoir-additive solutions. The reservoir solutions contained Tris-HCl 0.1 M pH 8.5, 30% PEG 4K and 0.2 M MgCl_2_ and the additive MnCl_2_ (0.1 M) (Hampton Research). The YtfE-E159L^Mn^ crystals were optimized by streak-seeding using the crystals of YtfE-E159L [40].

Needle-shaped crystals appeared a few minutes after plate set-up and grew to their maximal dimensions in 3–4 days. Crystals were harvested, immersed in a cryoprotectant solution (with the same composition as the reservoir solution supplemented with 25% (*v*/*v*) glycerol), flash frozen in liquid nitrogen, and sent to a synchrotron beamline for data collection. The YtfE^Mn^ data sets were collected at ALBA beamline XALOC [73] (Barcelona, Spain). The YtfE-E159L^Mn^ data set was recorded at beamline ID29 of the European Synchrotron Radiation Facility (ESRF, Grenoble, France). Fluorescence scans near the Mn and Fe K-absorption edges were performed at the XALOC beamline on a YtfE^Mn^ crystal and a YtfE^wt^ crystal, respectively. The data were analyzed using the CHOOCH program [74] (Figure 5) and used to select three energies for data collection from the same YtfE^Mn^ crystal, namely before the Mn K-edge, at the Mn K-edge peak, and at the Fe K-edge peak. Statistics for data collection and processing for the anomalous data are represented in Table 6.

The three YtfE^Mn^ and the single YtfE-E159L^Mn^ data sets were integrated and scaled with XDS [75] and autoPROC [76], analyzed with POINTLESS [77], and merged with AIMLESS [78]. The YtfE^Mn^ dataset collected at the Mn K-edge (peak Mn) was selected for structure determination and refinement. The YtfE^Mn^ and YtfE-E159L^Mn^ structures were solved by molecular replacement with PHASER [79] via the CCP4 Graphics User Interface [80,81] using the previously reported *E. coli* YtfE structure (PDB 7BHA) [40] as the search model. After an initial refinement with REFMAC [82], the models were improved by successive cycles of correction using COOT [83]. The other two YtfE^Mn^ datasets (before Mn K-edge and Fe K-edge peak) were phased via refinement with REFMAC against the YtfE^Mn^ (Mn peak) structural data. At this stage, anomalous difference maps were calculated with FFT via the CCP4 Graphics User Interface for the Mn K-edge and Fe K-edge datasets.

Refinement of the YtfE^Mn^ and YtfE-E159L^Mn^ structures continued with PHENIX [84] in five macrocycle steps, with the refinement of positional coordinates and individual isotropic atomic displacement parameters for all non-hydrogen atoms, occupancies, and using non-crystallographic symmetry restraints for the two independent molecules in the asymmetric unit. Hydrogen atoms were added to the structural models and included in the refinement in calculated positions. The examination and editing of the models between refinements were carried out with COOT against σ_A_-weighted 2|F_o_| − |F_c_| and |F_o_| − |F_c_| electron density maps. TLS (translation-liberation-screw) rigid body refinement of atomic displacement parameters was carried out for all structures, followed by refinement of individual isotropic B-factors. Two TLS groups were used for each chain, corresponding approximately to the two protein domains. Water molecules were added with PHENIX and verified with COOT. MOLPROBITY [46] was used to examine the model geometry together with the validation tools available in COOT. The structure’s figures were created using the PyMOL, Molecular Graphics System, Version 2.3.4 Open Source (The Pymol Molecular Graphics System, Version 2.0. Schrödinger LLC) [85].

The final atomic coordinates and experimental structure factors were deposited in the Worldwide Protein Data Bank [86] with the accession codes 7BE8 and 7OYI for the YtfE^Mn^ and YtfE-E159L^Mn^ structures, respectively. Data collection and refinement statistics are summarized in Table 3.

### 9.2. Fluorescence Measurements to Estimate Metal Binding Affinities

*E. coli* YtfE wild type and E159L proteins were expressed and purified as described previously [4] but in their apo-form, which was achieved by omitting the addition of iron to the growth medium. The iron content was determined by the TPTZ method [87]. Fluorescence spectra were collected at room temperature on a Cary Varian spectrofluorometer (Agilent). Apo-YtfE and apo-E159L variant protein solutions (5 µM), were prepared in 50 mM HEPES (pH 6.8) with 150 mM NaCl under reducing conditions (with 2 mM Tris (2-carboxyethyl)phosphine, TCEP). Samples were excited at 280 nm and fluorescence spectra were collected in the 290 to 450 nm range [49]. The maximum fluorescence intensity occurred at ~326 nm. MnCl_2_ and (NH_4_)_2_Fe(SO_4_)_2_ were used as metal sources. Solutions with different metal concentrations (2.5, 5, 10, 15, 20, 25 µM of Fe^2+^ or Mn^2+^) were prepared and incubated with each protein (at 5 µM) for 30 min, after which the fluorescence spectra were recorded. Metal-binding affinities were determined based on the fluorescence changes at 326 nm that occur upon the addition of the metal ion solution. Metal-binding affinities were calculated as metal association constants using GraphPad Prism 5 software.

## Figures and Tables

**Figure 1 molecules-27-04051-f001:**
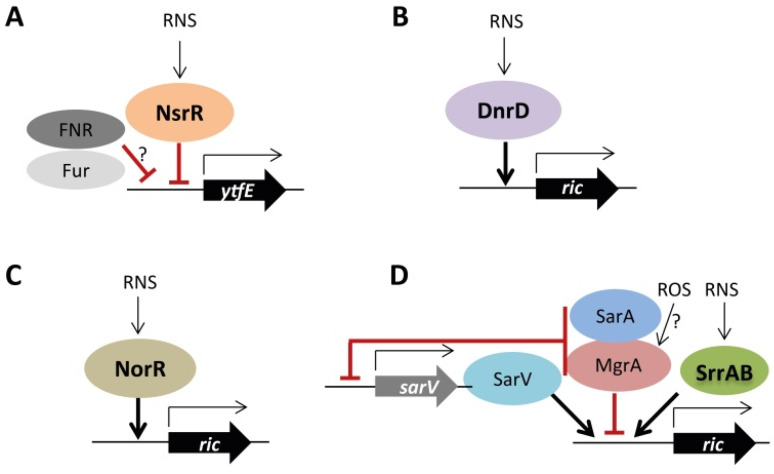
Regulation of *ric* in bacteria. (**A**)—*E. coli* YtfE is negatively regulated by NsrR and possibly indirectly repressed by FNR and Fur. (**B**)—*P. stutzeri ric* is under the positive control of DnrD. (**C**)—*R. eutropha ric* is regulated by NorR. (**D**)—*S. aureus ric* is regulated by the two-component system SrrAB in response to RNS. Adapted from [27].

**Figure 2 molecules-27-04051-f002:**
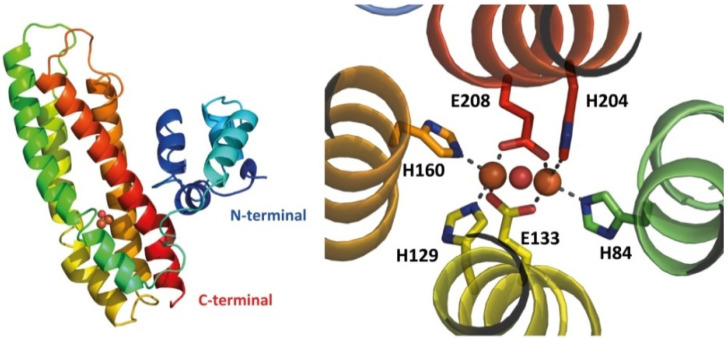
Structure of *E. coli* YtfE. The left panel shows a carton diagram of the YtfE monomer, rainbow-colored from the N-terminal (in dark blue) to the C-terminal (in red). The right panel displays a close-up view of the di-iron site, with the protein backbone shown in cartoon representation and the ligand residues depicted as sticks, with the iron atoms represented as orange spheres and the oxygen in the oxo-bridge as a red sphere (PDB entry 7BHA).

**Figure 3 molecules-27-04051-f003:**
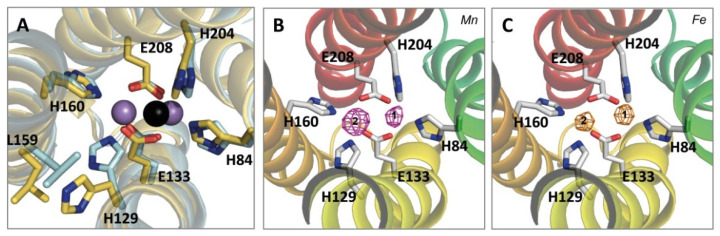
Structures of *E. coli* YtfE- E159L/YtfE-159L^Mn^ and YtfE^Mn^. (**A**)—Superposition of YtfE-E159L (yellow cartoon with the only iron atom shown as a black sphere) with YtfE-E159L^Mn^ (cyan cartoon with the metal sites occupied by Mn atoms represented as violet spheres). Figure shows that residues L159 and H129 have their side-chain positions shifted when Mn occupies the center. (**B**,**C**)—Anomalous difference maps of wild type YtfE co-crystallized with an excess of Mn. View of the anomalous difference map peaks drawn at the 8σ level in the di-metal site calculated from anomalous differences obtained at the Mn K-edge (1.8917 Å) ((**B**), purple mesh) and at the Fe K-edge (1.7313 Å) ((**C**), orange mesh). Ligand residues are represented as sticks and the monomer is shown in cartoon representation, colored in rainbow colors from blue (N-terminal) to red (C-terminal).

**Figure 4 molecules-27-04051-f004:**
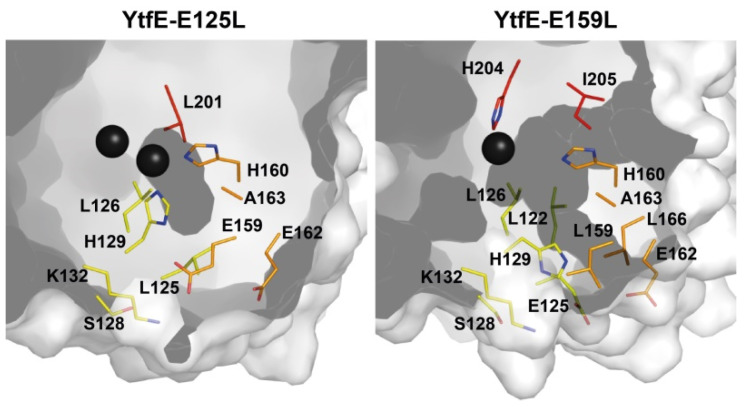
Structural comparison of the iron channel region in YtfE-E125L and YtfE-E159L. In YtfE-E125L a small cavity is present nearby the di-iron center, but the channel formation becomes blocked when leucine replaces E125. In YtfE-E159L, a channel entrance formed by residues E162, L166, and E125, still connects the external surface to the mononuclear iron center. Adapted from [40].

**Figure 5 molecules-27-04051-f005:**
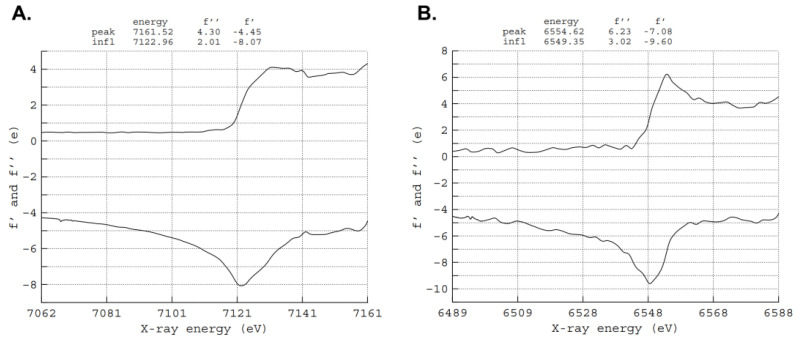
Analyses of fluorescence scans near the K-absorption edges of Fe and Mn using CHOOCH (**A**) Fe edge from a crystal of YtfE^wt^; (**B**) Mn edge from a crystal of YtfE^Mn^.

**Table 1 molecules-27-04051-t001:** Studied microbial RIC proteins.

Organism		Molecular Mass (kDa)	Fe Atoms	Oligomeric Form	Identity * (%)	References
*Escherichia coli*	YtfE	25	2	Monomer/dimer	-	[4]
*Staphylococcus aureus*	RIC	25	2	Monomer/dimer	25	UR
*Trichomonas vaginalis*	RIC 1	28	2	Nd	26	[7]
	RIC 2	28	0	Nd	49	
*Neisseria gonorrhoeae*	RIC	18	2	Monomer/dimer	31	UR

* Amino acid sequence identity with *E. coli* YtfE. Nd—not determined. UR—our unpublished results.

**Table 2 molecules-27-04051-t002:** Distances (Å) between the atoms of coordinating ligands relative to the di-metal site in YtfE-E159L^Mn^ mutant, and comparison with equivalent distances observed in di-iron containing YtfE wild type and E159L variant.

	YtfE-E159L^Mn^		YtfE *			YtfE-E159L *		
Mn1	H84 N^ε2^	2.16	Fe1	H84 N^ε2^	2.19	Fe1	H84 N^ε2^	2.72
	H204 N^ε2^	2.29		H204 N^ε2^	2.16		H204 N^ε2^	2.35
	E133 O^ε1^	2.20		E133 O^ε1^	2.11		E133 O^ε21^	1.97
	E208 O^ε2^	2.05		E208 O^ε2^	2.06		E133 O^ε2^	2.63
	O	2.48		O	2.02		E208 O^ε1^	2.53
Mn2	H129 N^ε2^	2.22	Fe2	H129 N^ε2^	2.13		E208 O^ε2^	1.91
	H160 N^ε2^	2.21		H160 N^ε2^	2.16		H_2_O	3.17
	E133 O^ε2^	2.05		E133 O^ε2^	2.00			
	E208 O^ε1^	2.21		E208 Oε1	2.07			
	O oxo-bridge	2.47		O oxo-bridge	2.05			

* From [40].

**Table 3 molecules-27-04051-t003:** Data collection, processing, and refinement statistics for the crystal structures of *E. coli* YtfE wild type and E159L proteins with manganese in the metal site.

	YtfE^Mn-pkMn^	YtfE-E159L^Mn^
Data Collection and Processing Statistics		
Beamline	ALBA XALOC	ESRF ID29
Detector	Dectris Pilatus 6M	Dectris Pilatus 6M
Wavelength (Å)	1.8917	0.9762
Space group	*P*2_1_	*P*2_1_
Unit-cell parameters (Å, °)	a = 60.14, b = 50.11, c = 88.22, β = 100.30	a = 59.82, b = 49.67, c = 88.09, β = 100.48
Data Processing	XDS/POINTLESS/AIMLESS	XDS/POINTLESS/AIMLESS
Resolution range (Å)	59.18–2.02 (2.05–2.02)	53.39–1.86 (1.89–1.86)
No. of observations	158206 (771)	142316 (7402)
Unique reflections	26890 (227)	43000 (2139)
<I/σ(I)>	10.5 (1.1)	13.0 (2.2)
R_merge_ ^a^	0.102 (1.149)	0.048 (0.563)
R_meas_ ^b^	0.122 (1.617)	0.058 (0.664)
R_pim_ ^c^	0.066 (1.137)	0.032 (0.350)
CC ^1/2 d^	0.996 (0.395)	0.999 (0.787)
Completeness (%)	78.3 (13.2)	99.3 (99.9)
Multiplicity	5.9 (3.4)	3.3 (3.5)
Wilson B-factor (Å^2^)	36.8	31.2
No. of molecules in a.u.	2	2
V_m_ (Å^3^ Da^−1^)	2.64	2.60
Estimated solvent content (%)	53.5	52.6
Structure refinement statistics		
Resolution limits (Å)	59.18—2.02 (2.09–2.02)	53.39–1.86 (1.90–1.86)
R-factor ^e^	0.2386 (0.3691)	0.1810 (0.3170)
nr. Reflections	26758 (590)	40900 (4092)
Free R-factor ^f^	0.2870 (0.3947)	0.2047 (0.3371)
nr. Reflections	1268 (30)	2089 (198)
Coordinate error estimate (Å)	0.28	0.25
Model completeness and composition:		
Omitted regions	1A, 1B–4B	1A, 1B
Nº molecules in asymmetric unit	2	2
Non-hydrogen protein atoms	3446	3738
Solvent molecules	100	227
Mean B values (Å^2^) ^g^:		
Protein chain A	40.6	34.9
Protein chain B	71.5	69.0
Solvent molecules	44.8	42.9
Model r.m.s. deviations from ideality:		
Bond lengths (Å)	0.001	0.008
Bond angles (°)	0.551	0.790
Chiral centers (Å^3^)	0.033	0.044
Planar groups (Å)	0.002	0.006
Model validation ^h^:		
% Ramachandran outliers	0	0
% Ramachandran favored	96.98	97.24
% Rotamer outliers	0.27	0
C^β^ outliers	0	0
Clash score	1.16	1.85
PDB accession ID	7BE8	7OYI

^a^ Rmerge = Σhkl Σi|Ii(hkl) − <I(hkl)>|/Σhkl Σi Ii (hkl), where Ii(hkl) is the observed intensity and <I(hkl)> is the average intensity of multiple observations from symmetry-related reflections [42]. ^b^ Rmeas = Σhkl [N/(N(hkl) − 1)]1/2 Σi|Ii(hkl) − <I(hkl)>|/Σhkl Σi Ii (hkl), where N(hkl) is the data multiplicity, Ii(hkl) is the observed intensity and <I(hkl)> is the average intensity of multiple observations from symmetry-related reflections. It is an indicator of the agreement between symmetry related observations [43]. ^c^ Rpim = Σhkl [1/(N(hkl) − 1)]1/2 Σi|Ii(hkl) − <I(hkl)>|/Σhkl Σi Ii (hkl), where N(hkl) is the data multiplicity, Ii(hkl) is the observed intensity and <I(hkl)> is the average intensity of multiple observations from symmetry-related reflections. It is an indicator of the precision of the final merged and averaged data set [44]. ^d^ CC1/2 = Correlation coefficient between intensities from random half-datasets [45]. ^e^ Rfactor = Σ|Fobs−Fcalc|/Σ Fobs, where Fobs and Fcalc are the amplitudes of the observed and the model calculated structure factors, respectively. It is a measure of the agreement between the experimental X-ray diffraction data and the crystallographic model. ^f^ cross-validation R-factor computed from a randomly chosen subset of 5% of the total number of reflections that was not used in the refinement. ^g^ Calculated from isotropic or equivalent isotropic B-values. ^h^ Calculated with MOLPROBITY [46]. Values in parenthesis are for the highest resolution shell.

**Table 5 molecules-27-04051-t005:** Distribution of RICs in microorganisms.

Class	Microorganism	RIC	Isc	Suf
Gammaproteobacteria	*Escherichia coli*	x	x	x
	*Salmonella enterica*	x	x	x
	*Shigella flexneri*	x	x	x
	*Klebsiella pneumoniae*	x	x	x
	*Yersinia pestis*	x	x	x
	*Yersinia pseudotuberculosis*	x	x	x
	*Pseudomonas aeruginosa*	x	x	
	*Haemophilus influenza*	x	x	
Actinomycetes	*Mycobacterium tuberculosis*	x		x
Betaproteobacteria	*Neisseria meningitidis*	x	x	
	*Neisseria gonorrhoea*	x	x	
Clostridia	*Clostridium difficile*	x	x	x
	*Clostridium perfringens*	x	x	x
Bacilli	*Bacillus subtilis*	x		x
	*Bacillus anthracis*	x		x
	*Listeria monocytogenes*	x		x
	*Staphylococcus aureus*	x		x
	*Staphylococcus epidermidis*	x		x
	*Streptococcus pneumoniae*	x		x
Zoomastigoporae (Eukaryote)	*Trichomonas vaginalis*	x	x	

**Table 6 molecules-27-04051-t006:** Data collection and processing statistics for the crystal structures of *E. coli* YtfE wild type with manganese in the metal site YtfE^Mn^, collected before the Mn K-edge (YtfE^Mn-bfMn^) and at the Fe K-edge peak (YtfE^Mn-pkFe^).

	YtfE^Mn-bfMn^	YtfE^Mn-pkFe^
Beamline	ALBA XALOC	
Detector	Pilatus 6M	
Wavelength (Å)	1.9016	1.7316
Space group	*P*2_1_	*P*2_1_
Unit-cell parameters (Å, °)	a = 60.15, b = 50.32, c = 88.26, β = 100.47	a = 60.26, b = 50.11, c = 88.44, β = 100.23
Data Processing	XDS/POINTLESS/AIMLESS	XDS/POINTLESS/AIMLESS
Resolution range (Å)	86.76–2.10 (2.14–2.10)	87.04–2.29 (2.33–2.29)
No. of observations	143929 (914)	143038 (5259)
Unique reflections	24438 (263)	23203 (1053)
<I/σ(I)>	11.6 (1.2)	6.4 (2.1)
R_merge_ ^a^	0.082 (0.823)	0.334 (2.734)
R_meas_ ^b^	0.090 (0.969)	0.367 (3.085)
R_pim_ ^c^	0.035 (0.504)	0.147 (1.376)
CC ^1/2 d^	0.997 (0.716)	0.997 (0.329)
Completeness (%)	79.9 (17.5)	98.3 (90.5)
Multiplicity	5.9 (3.5)	6.2 (5.0)
Wilson B-factor (Å^2^)	49.5	48.0
No. of molecules in a.u.	2	2
V_m_ (Å^3^ Da^−1^)	2.63	2.63
Estimated solvent content (%)	53.2	53.2

^a^ Rmerge = Σhkl Σi|Ii(hkl) − <I(hkl)>|/Σhkl Σi Ii (hkl), where Ii(hkl) is the observed intensity and <I(hkl)> is the average intensity of multiple observations from symmetry-related reflections [42]. ^b^ Rmeas = Σhkl [N/(N(hkl) − 1)]1/2 Σi|Ii(hkl) − <I(hkl)>|/Σhkl Σi Ii (hkl), where N(hkl) is the data multiplicity, Ii(hkl) is the observed intensity and <I(hkl)> is the average intensity of multiple observations from symmetry-related reflections. It is an indicator of the agreement between symmetry-related observations [43]. ^c^ Rpim = Σhkl [1/(N(hkl) − 1)]1/2 Σi|Ii(hkl) − <I(hkl)>|/Σhkl Σi Ii (hkl), where N(hkl) is the data multiplicity, Ii(hkl) is the observed intensity and <I(hkl)> is the average intensity of multiple observations from symmetry-related reflections. It is an indicator of the precision of the final merged and averaged data set [44]. ^d^ CC1/2 = Correlation coefficient between intensities from random half-datasets [45].

## Data Availability

The final atomic coordinates and experimental structure factors were deposited in the Worldwide Protein Data Bank with the accession codes 7BE8 and 7OYI for the YtfE^Mn^ and YtfE-E159L^Mn^ structures, respectively.

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
