# Peer review of "Repair of Iron Center Proteins—A Different Class of Hemerythrin-like Proteins"

_molecules, 2022, doi:10.3390/molecules27134051_

Round 1
Reviewer 1 Report
Dear authors, in order to improve this work, you have to take in your consideration these criticisms
- Title: the title should be rephrased, it isn’t recommended to use abbreviation in the tittle “RICs “
- Keywords: they are large number and most keywords aren’t present in the abstract please rephrase as “X-ray and crystallography”
- Add the abbreviation of “Repair of Iron Center proteins” and “Escherichia coli” to be used throughout the abstract and the manuscript
- The second sentence need rephrasing “First identified in Escherichia coli, YtfE and all other RICs contain a binuclear nonheme iron site in a four-helix bundle fold, two basic features of hemerythrin‐like proteins”.
- In line 41, please used the abbreviation of Escherichia coli and throughout the manuscript
- Line 45, it is incorrect sentence “ coli YtfE homologues are broadly spread in bacterial phyla” did you mean the protein “YTFE”
- Throughout the manuscript, revise the scientific written of the gene “ lowercase italic” and the protein “upper case non italic” for example in line 33 and 45, YtfE must be upper case non italic “YTFE”
- Can you add paragraphs about the possible compounds which can inactivate the RICS and can be used or increased the activities of antimicrobial agents
- Please add at the end of the review the limitations: what can’t this review tell us?
- Throughout the manuscript, Extensive editing of English language and style required
- Conclusion : It must be rephrased; conclusion section must provide us with the applied implication of your results in concise manner
Author Response
Dear Reviewer,
We thank your suggestions that were all taken into consideration. Title was modified. Keywords and abbreviations were revised. English revision of the manuscript was also done.
As for the scientific written of the protein we used what has been always used for this protein by us and other authors, i.e. YtfE. We verified and confirmed that all genes are written as lowercase italic.
To our best knowledge so far, there are no studies about possible compounds that could inactivate the RICs and could be used or increased the activities of antimicrobial agents.
We have further reinforced in the conclusion the still unknown issues about YtfE in the sentence – “studies are still needed to fully understand the physiological conditions that trigger iron release from YtfE and the systems that profit from this iron source”
The applied implication of the results are indicated in the sentence—“ role of RICs as iron donor proteins for the assembly of iron-sulfur proteins, which is a novel function for hemerythrin-like proteins.”-
A revised version of the manuscript is submitted.

Reviewer 2 Report
The core purpose of writing a paper is "RICs - a different class of hemerythrin-like proteins". The networking of protein interactions involving YtfE is also described and integrated in the proposed physiological role as iron donor for reassembling of iron-sulfur centers damaged by oxidative and nitrosative stresses.
Additionally, the authors present novel structural data for E. coli YtfE harboring a di-manganese site and the protein`s affinity for this metal.
This is a well-written paper containing interesting results which merit publication.
Author Response
Dear Reviewer,
We thank your positive opinion on this review.
Reviewer 3 Report
Dear editor and authors:
This article focused on the recent advances in the biochemical, spectroscopic and structural data of first identified bacterial YtfE. Generally, the manuscript is well-organized and the summary of YtfE is valuable for the further understanding of its protection mechanism. However, there are still some contents that need to be supplemented and revised. Firstly, is there any application of YtfE to modify pathogenic bacteria? In addition, the regulation of YtfE should be described in a figure to let readers easily understand. Thirdly, the effect of other metal ions and chemicals on YtfE should also be introduced.
Author Response
Dear Reviewer,
We thank your suggestions that were all taken into consideration.
To our best knowledge there are no studies on any application to modify pathogenic bacteria, and on the effect of other metal ions and chemicals on YtfE.
Following your suggestion, we introduced a new Figure 1 that describes data on the regulation of RICs in the bacteria that were so far analysed.
A revised version of the manuscript that includes English revision is submitted.

Round 2
Reviewer 1 Report
the authors added more information and all criticisms were managed, I believe that the manuscript has been considerably improved and it is suitable for publication in this version
Author Response
Dear Editor
please find the revised manuscript with the new data included.
Best regards
Lígia Saraiva
